# Single-Nucleus Transcriptome Sequencing Unravels Physiological Differences in Holstein Cows Under Different Physiological States

**DOI:** 10.3390/genes16080931

**Published:** 2025-08-03

**Authors:** Peipei Li, Yaqiang Guo, Yanchun Bao, Caixia Shi, Lin Zhu, Mingjuan Gu, Risu Na, Wenguang Zhang

**Affiliations:** 1College of Animal Science, Inner Mongolia Agricultural University, Hohhot 010010, China; 17201561597@163.com (P.L.); gggyaqiang@163.com (Y.G.); byc107054@163.com (Y.B.); caixiashi@imau.edu.cn (C.S.); zhulinynacxhs@163.com (L.Z.); gmj0119@yeah.net (M.G.); 2Inner Mongolia Engineering Research Center of Genomic Big Data for Agriculture, Hohhot 010018, China

**Keywords:** snRNA-Seq, Holstein cow, ovary, pregnant, cellular atlas, cell communication

## Abstract

**Background:** Against the backdrop of the large-scale and intensive development of the livestock industry, enhancing the reproductive efficiency of cattle has become a crucial factor in industrial development. Holstein cows, as the most predominant dairy cattle breed globally, are characterized by high milk yield and excellent milk quality. However, their reproductive efficiency is comprehensively influenced by a variety of complex factors, and improving their reproductive performance faces numerous challenges. The ovary, as the core organ of the female reproductive system, plays a decisive role in embryonic development and pregnancy maintenance. It is not only the site where eggs are produced and developed but it also regulates the cow’s estrous cycle, ovulation process, and the establishment and maintenance of pregnancy by secreting various hormones. The normal functioning of the ovary is crucial for the smooth development of the embryo and the successful maintenance of pregnancy. **Methods:** Currently, traditional sequencing technologies have obvious limitations in deciphering ovarian function and reproductive regulatory mechanisms. To overcome the bottlenecks of traditional sequencing technologies, this study selected Holstein cows as the research subjects. Ovarian samples were collected from one pregnant and one non-pregnant Holstein cow, and single-nucleus transcriptome sequencing technology was used to conduct an in-depth study on the ovarian cells of Holstein cows. **Results:** By constructing a cell type-specific molecular atlas of the ovaries, nine different cell types were successfully identified. This study compared the proportions of ovarian cell types under different physiological states and found that the proportion of endothelial cells decreased during pregnancy, while the proportions of granulosa cells and luteal cells increased significantly. In terms of functional enrichment analysis, oocytes during both pregnancy and non-pregnancy play roles in the “cell cycle” and “homologous recombination” pathways. However, non-pregnant oocytes are also involved in the “progesterone-mediated oocyte maturation” pathway. Luteal cells during pregnancy mainly function in the “cortisol synthesis and secretion” and “ovarian steroidogenesis” pathways; non-pregnant luteal cells are mainly enriched in pathway processes such as the “AMPK signaling pathway”, “pyrimidine metabolism”, and “nucleotide metabolism”. Cell communication analysis reveals that there are 51 signaling pathways involved in the pregnant ovary, with endothelial cells, granulosa cells, and luteal cells serving as the core communication hubs. In the non-pregnant ovary, there are 48 pathways, and the interaction between endothelial cells and stromal cells is the dominant mode. **Conclusions:** This study provides new insights into the regulatory mechanisms of reproductive efficiency in Holstein cows. The differences in the proportions of ovarian cell types, functional pathways, and cell communication patterns under different physiological states, especially the increase in the proportions of granulosa cells and luteal cells during pregnancy and the specificity of related functional pathways, indicate that these cells play a crucial role in the reproductive process of cows. These findings also highlight the importance of ovarian cells in pathways such as “cell cycle”, “homologous recombination”, and “progesterone-mediated oocyte maturation”, as well as the cell communication mechanisms in regulating ovarian function and reproductive performance.

## 1. Introduction

In the reproductive system of female mammals, the hypothalamic–pituitary–ovarian axis plays a central regulatory role. The hypothalamus secretes a gonadotropin-releasing hormone (GnRH), which stimulates the pituitary gland to secrete a follicle-stimulating hormone (FSH) and luteinizing hormone (LH). These hormones act on the ovaries to regulate their growth, development, and function. The precise signal transmission and feedback regulation of this axis are crucial for maintaining the stable operation of female reproductive physiological processes. Among them, as an important executive organ in this axis, the ovary occupies a central position and is a crucial reproductive organ with multiple key physiological functions. This organ consists of follicles at different developmental stages. It is not only the sole source of oocytes but also the main site for the secretion of steroid sex hormones [1]. Many functions of the ovary, such as continuously providing oocytes with fertilization potential, precisely secreting reproduction-related hormones, and finely regulating the estrous cycle, are direct determinants of the reproductive efficiency of female animals [2]. As the central regulatory organ of the reproductive system, the ovary produces sex hormones to coordinate female secondary sexual characteristics, stores developmentally competent mature oocytes, releases mature oocytes for fertilization, and maintains pregnancy [3,4,5,6]. Moreover, the ovary is the most crucial and complex reproductive organ in females. Through ovarian cell types such as supporting granulosa cells (GCs) and follicular cells, it provides steroid sex hormones and mature oocytes, and maintains endocrine homeostasis and female fertility [7,8].

As the core organ of the female reproductive system, the functional state of the ovary directly determines the success rate of embryo development and pregnancy maintenance. The ovary has two key physiological functions. Firstly, it is responsible for regulating the development, differentiation, and release of oocytes, thus participating in the reproductive process [9]. Secondly, it is responsible for synthesizing and secreting sex steroid hormones, including estrogens, progestogens, and androgens. These hormones play a crucial role in maintaining follicular development, ensuring fertility, regulating the normal rhythm of the menstrual/estrous cycle, and supporting pregnancy [10]. During the process of cyclic changes, there are follicles at different developmental stages in the ovary. After ovulation, depending on the species, one or more corpora lutea may form. The process by which the most primitive follicles (primordial follicles) gradually develop into pre-ovulatory follicles is called folliculogenesis [11]. Notably, the realization of ovarian function highly depends on cell heterogeneity. The differences in the transcriptomic characteristics between granulosa cells (EGR4high/FSTlow) and theca cells (AKIRIN1high/LHP high) in antral follicles determine the efficiency of steroid synthesis and the quality of oocyte maturation [12].

The ovary consists of numerous heterogeneous cell types, including oocytes, granulosa cells, stromal cells, endothelial cells, and immune cells [13]. As the fundamental functional unit of the ovary, the follicle is primarily composed of an oocyte, granulosa cells, and theca cells [14]. Granulosa cells are the main component of the follicular wall and play a crucial role in regulating follicular development [15,16]. They not only provide nutrients for the oocyte but also exert significant regulatory effects on the growth and development of the follicle through the secretion of hormones, cytokines, and regulatory proteins. Importantly, abnormal function of granulosa cells is closely associated with reproductive system diseases [17,18,19]. Therefore, deciphering the molecular basis of GC development is of great significance for the diagnosis and treatment of ovarian diseases. Moreover, previous studies have primarily focused on the impacts of granulosa cells and oocytes on ovarian function [20,21]. Nowadays, there is a growing recognition of the functional roles of ovarian somatic cells, such as endothelial cells, stromal cells, pericytes, and immune cells, in follicular development [22,23].

However, traditional bulk RNA sequencing (bulk RNA-seq) can only obtain the average expression profiles of tissues and is unable to decipher the specific molecular mechanisms of different cell types [24]. Thanks to the development of high-throughput single-cell sequencing technology, more subtypes of ovarian cells have been discovered, and the heterogeneity of ovarian cells has been further investigated. Mapping the ovarian cell atlas is crucial for understanding folliculogenesis, ovulation, and pregnancy maintenance. To date, the cellular and molecular characteristics within the ovaries of some species have been revealed through single-cell RNA sequencing technology. Comprehensive cell atlases have been constructed for humans [25,26], monkeys [27,28], Drosophila [29,30,31], and some fish species [32,33,34]. In these reports, the ovarian cells of these species have been classified into different cell types and characterized based on their specifically expressed genes. In addition, the breakthrough in single-nucleus RNA sequencing (snRNA-seq) technology offers a new avenue for overcoming the aforementioned limitations. This technology performs transcriptome analysis by isolating individual cell nuclei, thereby circumventing the data biases caused by differences in cell size during the isolation of intact cells [35]. Its reliability has been verified in studies such as the classification of human testicular spermatogenic cells [36] and the mechanisms underlying premature ovarian failure [37]. Nevertheless, there are still significant gaps in the cell atlas of Holstein cow ovaries. Most existing studies focus on human pathological models and lack a systematic analysis of the normal physiological state of cattle [38]. The differences in the cellular interaction networks between the ovaries during pregnancy and non-pregnancy have not been clearly defined. However, such dynamic changes are precisely the key windows for regulating reproductive efficiency [12]. In the field of bovine ovarian research, the cell types and their molecular signals that regulate ovarian function remain unclear. Understanding the overall cell types in the ovaries under different physiological conditions is crucial for elucidating the reproductive adaptation mechanisms of cattle, a large-sized species. Therefore, constructing a cell atlas of the bovine ovary, analyzing the intercellular communication signals, and then elaborating on its reproductive mechanisms have become highly valuable research directions in this field. In view of this, this study selected Holstein cows as the research model and for the first time employed single-nucleus RNA sequencing (snRNA-seq) technology. The aim is to construct a cell type-specific molecular atlas of Holstein cow ovaries, with the hope of providing an important basis for in-depth exploration of the functional regulatory mechanisms and reproductive mechanisms of bovine ovaries.

## 2. Materials and Methods

### 2.1. Ovarian Sample Collection and Suspension Preparation

In this experiment, the bovine ovarian tissues were collected from Holstein cows that had been slaughtered at abattoirs around Hohhot, Inner Mongolia. Among them, for one cow, when the ovaries were collected, an obvious fully formed fetus was found in its uterus. The left and right ovaries of this pregnant cow were labeled as O25xh and O25d, respectively. For another cow, after the ovaries had been collected, obvious corpora lutea were observed in them. The left and right ovaries of this non-pregnant cow were labeled as O19x and O19d, respectively. Both of these two cows were 4–5 years old. After the collection of ovarian tissues, they were first rinsed multiple times with phosphate-buffered saline (PBS) to eliminate surface bleeding. Subsequently, a scalpel was used to cut the tissues into small pieces approximately 2–4 mm^3^ in size. Immediately after that, the collected tissues were immersed in pre-cooled tissue preservation solution and transported to the laboratory for subsequent processing within 4 h under 4 °C conditions.

Before starting the experiment, the Singulator 100 instrument was pre-cooled. Meanwhile, digestive solution LB containing 1% BSA was prepared (approximately 2.5 mL was consumed for each sample, and it was prepared fresh according to the experimental requirements). DPBS was added to the reagent bottle supplied with the Singulator 100, and then the reagent bottle was connected to the nuclear suspension reagent line (NSR). Subsequently, the slightly trimmed sample was placed into the inner slot of the cartridge, and 2 mL of digestive solution LB was added. Then, the standard program for single-nucleus preparation on the Singulator 100 was selected and run. After the instrument finished running, the cartridge was removed, the aluminum film of the cartridge was pierced, and the nuclear suspension was aspirated and transferred to a 15 mL centrifuge tube. The volume was topped up to 8–10 mL with DPBS, and then it was centrifuged at 500× *g* for 5 min at 4 °C. After centrifugation, the supernatant was slowly aspirated, leaving the pellet. Then, 300 μL of digestive solution containing 1% BSA was added to the pellet, and the nuclei was gently pipetted up and down to resuspend them thoroughly. Subsequently, the resuspended nuclei were transferred to a new 2 mL Eppendorf tube (EP). Next, 300 μL of tissue dissociation density gradient solution 1 (TD1) was added and pipetted thoroughly again to ensure uniform mixing. Then, 600 μL of tissue dissociation density-gradient solution 2 (TD2) was aspirated, the pipette tip was inserted to the bottom of the EP tube, and the TD2 solution was slowly added to form a solution layer. Similarly, 600 μL of tissue dissociation density-gradient solution 3 (TD3) was aspirated, the pipette tip was inserted to the bottom of the EP tube, and the TD3 solution was slowly added to form another solution layer. After completing the solution layering, gradient density centrifugation was performed at 3000× *g* for 20 min at 4 °C. After centrifugation was completed, the top 600 μL of the supernatant was removed in sequence. Approximately 150 μL of nuclei was aspirated from the interface between the TD2 and TD3 solutions and transferred to a new 1.5 mL EP tube. Then, 1 mL of BR solution was added and pipetted thoroughly to mix well, and the solution was filtered through a 30 μm cell strainer and centrifuged at 500× *g* for 5 min at 4 °C. The supernatant was slowly aspirated again, leaving the pellet. Then, 500 μL of buffer reagent (BR) solution was added, pipetted to resuspend the nuclei, and then centrifuged at 500× *g* for another 5 min at 4 °C. The supernatant was removed and the pellet was retained. Then, 50 μL of BR solution was added to resuspend the nuclei. If no pellet was observable, 50 μL was left when removing the supernatant. Finally, various parameters were measured using a cell counter.

### 2.2. Single-Cell Library Construction and Sequencing

After the amplified products passed the quality inspection, the construction of the sequencing library began. Chemical methods were used to fragment the cDNA to a length of approximately 200–300 bp. Subsequently, end repair and A-tailing was performed on these cDNA fragments. Suitable-sized cDNA fragments were selected through fragment screening, and sequencing adapters were ligated. Then, specific sample indexes were introduced via PCR amplification, followed by a strict fragment screening process to obtain a single-cell sequencing library that met the requirements. After the library construction was completed, a systematic quality control protocol was adopted: The Qubit 3.0 fluorescence quantification system was used to perform preliminary quantification of the library, and the library was standardized to a working concentration of 1 ng/μL. The Agilent 2100 bioanalyzer (Agilent Technologies, Beijing, China)was used to detect the insert size distribution of the library to ensure that the fragment size remained stable within the optimized range of 200–500 bp. The StepOnePlus real-time fluorescent quantitative PCR system was used to accurately measure the effective concentration of the library. The effective concentration of all libraries met the quality control standard of ≥10 nM. After the above strict quality control processes, the qualified libraries were subjected to subsequent sequencing on the BGI high-throughput sequencing platform, using a paired-end 150 bp (PE150) sequencing strategy.

### 2.3. Single-Nucleus RNA Sequencing Data Processing

In this study, standard analysis was conducted on the raw sequencing FASTQ data. First, genome alignment and cell barcode identification were performed. Cellranger software (v7.1.0)was used to conduct quantitative analysis of unique molecular identifiers (UMIs), resulting in a gene feature-cell barcode expression matrix. R v4.2.2, Seurat v5.2.1, and harmony were utilized. The unstandardized bovine ovarian data were loaded into R using the Read10X function to construct a Seurat object, and then the merge function was used to combine and default normalize the data. Subsequently, the raw sequencing data were imported into Rstudio(v4.2.2), and a Seurat object was created using the Create Seurat Object function. Data quality control was carried out, filtering cells with a high proportion of mitochondrial genes, those with more than 10,000 or less than 200 detected genes, and cells with a total count of less than 500. After that, the NormalizeData function was used to eliminate the influence of sequencing depth differences. The FindVariableFeatures function was employed to identify highly expressed and highly variable genes. ScaleData was used for z-score standardization to prepare for principal component analysis (PCA) using the RunPCA function. Considering that the four ovarian tissue samples were obtained from the left and right ovaries of two cows in different physiological states, and were prepared and sequenced in four separate batches, the harmony algorithm was adopted. With “orig.ident” as the source of batch effects, the RunHarmony function was used to remove the batch effects from the Seurat object.

### 2.4. Dimensionality Reduction, Cell Clustering, and Marker Gene Selection

Principal component analysis (PCA) was a core method for the linear dimensionality reduction of data. Its principle was to map high-dimensional data from n dimensions to a low-dimensional space of k dimensions, generating principal components to capture the main variation information of the data. Significance analysis of the principal components was carried out through the JackStraw program and permutation tests. The JackStrawPlot function was used to screen significant principal components, and the ElbowPlot function was used to display the cumulative variance contribution rate to determine the number of principal components. In cell clustering analysis, the Seurat (v5.2.1) software package adopted a graph-based clustering algorithm. First, the FindNeighbors function was used to calculate the neighborhood relationship of cells and construct a k-nearest neighbor graph to determine the optimal dimensional range. The resolution parameter of the FindClusters function adjusted the clustering fineness. This study used two non-linear dimensionality reduction techniques, UMAP and tSNE, to map cells to a low-dimensional space, and the DimPlot function was used to visualize the results. In terms of differential expression analysis, Seurat’s FindMarkers was used to compare differentially expressed genes (DEGs) between two specific clusters, and FindAllMarkers analyzed DEGs of all clusters at once. In this study, it was used to screen DEGs for each cluster. Subsequently, functions such as VlnPlot, FeaturePlot, DotPlot, and DoHeatmap were used to visualize gene expression. Functional annotation of cell sub-populations was performed based on the expression profiles of known tissue-specific marker genes. Since automatic annotation tools such as SingleR and celldex have limited support for species like cattle, this study performed manual annotation based on classic marker genes and the relevant literature. The VlnPlot and DotPlot functions were used to verify the expression of marker genes to determine cell types.

### 2.5. Gene Ontology and Kyoto Encyclopedia of Genes and Genomes

In Rstudio, the “clusterProfiler” and “org.Bt.eg.db” packages from Bioconductor were employed to conduct gene ontology functional enrichment analysis and Kyoto Encyclopedia of Genes and Genomes pathway enrichment analysis on the top 300 differentially expressed genes (DEGs) within each cell cluster.

### 2.6. Cell Communication Analysis

Cell–cell communication (CCC) refers to the interactive process of regulation between cells through biochemical signals. This process can regulate the life processes of individual cells and the relationships between cells. Conducting cell-to-cell communication analysis is of crucial significance for deciphering the complex interaction patterns between cells and clarifying the topological structure and functional logic of the cell communication network. Based on this, the CellChat tool was used to conduct a systematic intercellular communication analysis on single-cell data. First, the CellChat object was created using the KYJ_cellchat function, and the cells were grouped based on cell type annotation. Subsequently, the CellChatDB database was used to identify significantly over-expressed ligand–receptor pairs, and the communication probability between cells and the activity of signaling pathways were calculated.

### 2.7. Construction of the Developmental Trajectories of Granulosa Cells and Luteal Cells

We performed pseudotime trajectory analysis using the R package monocle. Granulosa cells and oocytes were analyzed separately according to their transcriptional dynamic characteristics. The Monocle object was constructed from the Seurat object. The DDRTree algorithm was used to reduce the dimensionality of the data, and the developmental trajectories were reconstructed in the pseudotime space. Highly variable genes identified from each cell population were used as inputs for trajectory visualization. The plot_pseudotime_heatmap function was used to display a heatmap of the dynamic changes in gene expression along the pseudotime axis. The trajectory plot was drawn using the plot_cell_trajectory function and colored according to the cell state, cluster, or pseudotime value to show the pattern of the developmental process. In addition, the plot_genes_in_pseudotime function was used to visualize the temporal expression trends of selected key genes, revealing their dynamic changes at different developmental stages.

## 3. Results

### 3.1. Data Integration, Quality Control, and Analysis

To investigate the cellular functional changes in the ovaries of Holstein cows, we conducted single-nucleus RNA sequencing (snRNA-seq) on ovaries in different physiological states (pregnancy (P): O25d and O25x; non-pregnancy (NP): O19d and O19x). We dissected the entire left and right ovaries, dissociated them into single cells, and then extracted the nuclei for snRNA-seq. After sequencing, the number of cells obtained from each sample ranged from 6023 to 7233, and the median number of genes per cell was between 1501 and 1635 (Appendix A). Subsequently, we performed quality control on the single-cell expression matrices of the four ovarian tissue samples using the Seurat package in the R programming language. To evaluate the data quality, we represented the gene distribution in cells of each sample by nFeature_RNA and the unique molecular identifier (UMI) distribution by nCount_RNA. These parameters were visualized, and cells with a gene count of less than 5000 and a mitochondrial gene proportion of less than 5% were retained (Appendix A). We used Harmony for the integrated analysis of the four datasets (Appendix A). After a series of quality control and data processing steps, a total of 26,159 cells expressing 21,525 genes were ultimately retained for subsequent analysis.

### 3.2. Cell Clustering and Marker Gene Selection

Based on the sequencing data, we employed a Seurat-based workflow for cell clustering. Through uniform manifold approximation and projection (UMAP) analysis, a total of 27 clusters were identified (Figure 1A). Differential gene expression analysis was conducted among these 27 clusters to characterize the various cell populations in the dataset. The expression specificity of the top five differentially expressed genes (DEGs) in each cluster was presented in a heatmap (Figure 1C). The expression levels and percentages of representative genes in different clusters were visualized in the dot matrix, indicating that each cluster had its own specifically expressed genes (Figure 1B).

### 3.3. Ovarian Somatic and Germ Cell Type Annotation

Although the SingleR package provides a method for automatic annotation of single-cell RNA (scRNA) data, it is not suitable for the annotation of cell types in the ovaries of Holstein cows, as there have been no reports of a marker gene database specifically for bovine ovaries to date. Therefore, based on the annotation results obtained from SingleR and by referring to existing cell markers, the ovarian somatic cell types were identified. Given that extensive research has been conducted on single-cell analysis of human and mouse ovarian tissues, systematic cell type annotation was performed on all cell clusters based on the known expression profiles of cell type specific marker genes (Figure 1B). FBLN1 [39], COL1A2, and PDGFRA [26] were differentially expressed in clusters 0, 3, 7, 16, and 19. Therefore, these clusters were annotated as stromal cells. KDR, CCL21, CCL14, CLDN5 [25], PECAM1 [40], and CD34 [41] showed relatively high expression levels in endothelial cells and are marker genes for endothelial cells. Consequently, clusters 1, 4, 5, 6, 10, 12, 14, 15, 18, 20, 22, 23, and 27 were annotated as endothelial cells. Clusters 9 and 25 were annotated as luteal cells due to the high expression of LHCGR [42] and CYP11A1. Given the relatively high expression levels of FSHR and FST [43] in clusters 2 and 8, these two clusters were annotated as granulosa cells. In cluster 24, CENPF and TOP2A showed high expression levels. CENPF is a cell cycle-associated protein that is mainly expressed during the G2/M phase and is involved in cell proliferation and mitosis, while TOP2A is closely related to cell proliferation in various cell types [44]. Therefore, cluster 24 was annotated as oocytes. Since ACTA2 and RGS5 [26] are classic marker genes for smooth muscle cells, clusters 11 and 17 were annotated as smooth muscle cells. CD53 and CCL5 [45] were specifically expressed in cluster 21, so this cluster was annotated as T/NK cells. CD163 [46] and VSIG4, which are classic marker genes for macrophages, exhibited relatively high expression levels in cluster 13. Therefore, cluster 13 was annotated as macrophages. KRT19 and KRT18 [46], as classic marker genes for epithelial cells, were specifically expressed in cluster 26. Hence, cluster 26 was annotated as epithelial cells.

Ultimately, nine cell types in the ovarian tissue were identified (Figure 2A,B), including oocytes (1.2%), endothelial cells (45.6%), epithelial cells (0.3%), granulosa cells (11.8%), luteal cells (5.3%), macrophages (2.5%), smooth muscle cells (6.4%), stromal cells (25.1%), and T/NK cells (1.7%). To further validate the cell types within the clusters, the expression specificity of the top 10 representative differentially expressed genes (DEGs) for each cell type was presented in a heatmap (Figure 2C). Most differentially expressed genes (DEGs) were specifically expressed in their own cell types. The expression levels of the 18 most representative cell type-specific genes, including PDGFRA, FBLN1, CD34, FSHR, CENPF, RGS5, CD163, KRT19, CCL5, LHCGR, CYP11A1, etc., were mapped on the uniform manifold approximation and projection (UMAP) plot (Figure 2E). Additionally, violin plots showed that marker genes had higher and more specific expression in their respective single cell types, which was consistent with the heatmap and feature plots, indicating that different cell types were identified only by their characteristic genes (Figure 2D). The expression scores and percentages of specifically expressed genes for different cell types were visualized in a dot plot (Figure 2F).

### 3.4. Differences in the Expression Profiles of Ovarian Somatic Cells Between the Pregnancy and Non-Pregnancy Periods

We compared the differences in the expression profiles of ovarian somatic cells in Holstein cows under different physiological conditions. In our study, the comparison of somatic cell expression profiles was based on cell types. There were differences in the proportions of cell types between the pregnancy and non-pregnancy periods. The proportions of granulosa cells and luteal cells were higher during pregnancy than in the non-pregnancy period, while the proportions of endothelial cells and immune cells were lower (Figure 2B and Figure 3A). Additionally, we collected the differentially expressed genes (top 300) of each cell type for gene ontology (GO) and Kyoto Encyclopedia of Genes and Genomes (KEGG) enrichment analyses, and visualized the top 10 DEGs of each cell type in a heatmap (Figure 3B). After the gene ontology (GO) and Kyoto Encyclopedia of Genes and Genomes (KEGG) enrichment analyses, bubble plots were generated based on the top five biological processes with the highest false discovery rate (FDR) enrichment for each cell type, which demonstrated the main biological functions of nine cell types (Figure 3C,D).

In the biological processes of the GO enrichment analysis (Figure 3D), the differentially expressed genes (DEGs) with high expression in endothelial cells during pregnancy were mainly involved in biological processes such as vasculature development, translation, peptide biosynthetic process, and amide biosynthetic process. In contrast, the DEGs in endothelial cells during the non-pregnancy period were primarily engaged in circulatory system development, blood vessel development, blood vessel morphogenesis, and other biological processes. The DEGs in oocytes during pregnancy and non-pregnancy were mainly enriched in biological processes including cell cycle process, chromosome segregation, and nuclear division.

In terms of KEGG pathways (Figure 3C), the highly expressed genes in endothelial cells during both pregnancy and the non-pregnancy period were enriched in pathways such as ribosome, Rap1 signaling pathway, and Ras signaling pathway. Oocytes during both pregnancy and the non-pregnancy period played roles in the cell cycle and homologous recombination pathways. However, oocytes during the non-pregnancy period were also involved in the progesterone-mediated oocyte maturation pathway. Luteal cells during pregnancy mainly functioned in the cortisol synthesis and secretion and ovarian steroidogenesis pathways. In contrast, luteal cells during the non-pregnancy period were mainly enriched in pathways such as the AMPK signaling pathway, pyrimidine metabolism, and nucleotide metabolism. The GO and KEGG enrichment analyses indicated that the biological processes and pathways in which each cell type was involved were consistent with its known biological functions.

### 3.5. Signal Transduction Crosstalk Among Various Cell Types in the Ovary Under Different Physiological Conditions

Intercellular communication plays a vital and indispensable role in maintaining the normal functions of complex tissues. Ovarian tissue demonstrates distinct functional manifestations during various physiological periods, and alterations in the complexity of intercellular communication might serve as the underlying key factor. To comprehensively elucidate the specific characteristics and intricate mechanisms of intercellular communication within ovarian tissue across different periods, it is particularly essential to conduct a thorough and meticulous analysis of the signal transduction crosstalk involving ligands, receptors, and their co-factors. By leveraging the R package CellChat, it is possible to more effectively explore the signal transduction crosstalk among the annotated cell types in ovarian tissue. The research results show that the ovarian somatic cell types during pregnancy revealed a total of 863 ligand–receptor pairs, which were further classified into 51 signaling pathways (Appendix A), including signaling pathways such as IGF, VEGF, PTPRM, NOTCH, and TGFβ. The ovarian cell types during the non-pregnancy period revealed a total of 847 ligand–receptor pairs, involving 48 signaling pathways (Appendix A), and these signaling pathways were largely consistent with those in the pregnant ovary.

Cell communication analysis of the cell types in the pregnant ovary revealed that both in terms of the number and strength of interactions, granulosa cells and endothelial cells played a central role in inter-ovarian cell communication. Stromal cells and luteal cells also showed prominent performance in terms of the number and strength of interactions, ranking second only to granulosa cells and endothelial cells (Figure 4A). During the non-pregnancy period, endothelial cells and stromal cells exhibited strong signaling connections (Figure 4A). Communication between granulosa cells, epithelial cells, and other cell populations via the VEGF and ANGPT pathways occurred exclusively during pregnancy (Figure 4B, Appendix A). Additionally, network centrality analysis of the VEGF and ANGPT pathways during the non-pregnancy period revealed that endothelial cells were the primary sources of these ligands, suggesting predominantly autocrine interactions (Figure 4C). The NOTCH and IGF signaling pathways indicated that endothelial cells and stromal cells were the main sources targeting endothelial cells (Appendix A). During pregnancy, the VEGF pathway indicated that granulosa cells were the primary source targeting endothelial cells. The ANGPT pathway showed that granulosa cells and endothelial cells were the main sources of ANGPT ligands targeting endothelial cells, suggesting autocrine and paracrine interactions (Figure 4C). Similarly, the NOTCH signaling pathway revealed that endothelial cells and luteal cells were the main sources targeting stromal cells, and the IGF signaling pathway showed that endothelial cells were the main sources targeting stromal cells, endothelial cells, and granulosa cells (Appendix A). Signal pathway analysis revealed that communication among cell populations was more frequent during pregnancy than in the non-pregnancy period. These findings underscore the intricate intercellular communication within ovarian cells of Holstein cows under different physiological states. There are significant differences in the cell types that act as the primary sources of ligands in each signaling pathway between the pregnancy and non-pregnancy periods. The same signaling pathway involves different cells and operates in different modes at different times. This complex communication network is of paramount importance for maintaining the normal physiological functions of the ovaries and adapting to the physiological demands of pregnancy and non-pregnancy. It is closely associated with physiological processes such as follicular development, corpus luteum formation and maintenance, and hormone secretion.

Subsequently, we conducted an analysis of communication patterns. During pregnancy, seven distinct patterns were identified in the incoming target cells, and five different patterns were identified in the outgoing secreting cells (Figure 4D). The incoming signal patterns in the target cells showed that endothelial cells exhibited Pattern 1, involving 18 pathways, including CLDN, CDH5, CALCR, etc. Granulosa cells exhibited Pattern 2, involving 10 pathways, including ncWNT, VISTA, CADM, etc. The outgoing signals from granulosa cells and oocytes presented Pattern 2, involving five pathways, including CADM, CALCR, CDH, VEGF, and NRG. During the non-pregnancy period, five incoming and four outgoing signal patterns were identified in target cells and secreting cells, respectively (Figure 4E). The incoming and outgoing signal patterns in target cells and secreting cells showed that smooth muscle cells and oocytes exhibited Pattern 3 and Pattern 4, respectively, both involving five pathways. The incoming and outgoing pathways both included NT, NEGR, and PROS. Meanwhile, the incoming and outgoing signal patterns of immune cells (macrophages and T/NK cells) were both Pattern 2, each involving six pathways. In addition, the incoming signals from granulosa cells and luteal cells exhibited Pattern 4, involving three pathways, including CDH, VISTA, and NRG. The outgoing signals from granulosa cells, luteal cells, and stromal cells exhibited Pattern 3, involving a total of four pathways: CALCR, THBS, PTN, and CDH.

### 3.6. Differential Analysis and Developmental Trajectory of Ovarian GCs Under Different Physiological States

In the CytoTRACE analysis, UMAP was utilized to visualize the differentiation potential of ovarian cells. As depicted in the figures, certain cell types, such as endothelial cells and stromal cells, displayed a relatively low degree of differentiation during pregnancy. In contrast, only endothelial cells exhibited a low level of differentiation during the non-pregnant state (Figure 5A, Appendix A). Additionally, granulosa cells showed lower differentiation potential during pregnancy compared with the non-pregnant period (Figure 5A, Appendix A). Given the significant disparities in the manifestation of differentiation potential between these two cell types, to gain a more in-depth understanding of their specific mechanisms of action and interrelationships in the physiological processes of the ovary, we will subsequently conduct a detailed differential analysis of granulosa cells and luteal cells.

In the differential analysis of granulosa cells between pregnant and non-pregnant ovaries, we found that, compared with non-pregnant state, 77 genes were upregulated in pregnant ovarian granulosa cells, including FST, ADM, AMH, INSL3, etc., and 333 genes were downregulated (Figure 5B). Subsequently, we conducted a Kyoto Encyclopedia of Genes and Genomes (KEGG) enrichment pathway analysis on these differentially expressed genes. The results indicated that these differentially expressed genes were extensively involved in multiple crucial signaling pathways and physiological processes. These included important signal transduction pathways such as the relaxin signaling pathway, insulin signaling pathway, AMP-activated protein kinase (AMPK) signaling pathway, and phosphatidylinositol 3-kinase (PI3K)-protein kinase B (Akt) signaling pathway, as well as physiological activities such as growth hormone synthesis, secretion, and action, and the oxytocin signaling pathway. In addition, cell differentiation is a complex process that progresses over time. To gain a deeper understanding of the dynamic developmental trajectories and gene expression regulatory patterns of granulosa cells and luteal cells over time in the physiological processes of the ovary, we will subsequently conduct a pseudotime analysis to uncover the dynamic developmental maps of these two cell types during ovarian physiological processes. For granulosa cells during pregnancy, the pseudotime analysis trajectory revealed two branching nodes, while three branching nodes were observed during the non-pregnant period (Figure 5D). We performed a heatmap analysis on cells in different developmental states and found that granulosa cells exhibited four patterns of gene expression levels during pregnancy and three patterns during the non-pregnant period. As can be seen from the heatmap (Figure 5E), the expression levels of the non-pregnant granulosa cell marker genes FSHR and FOXO1 were low at the initial stage of pseudotime and increased sharply at the later stage. In pregnant granulosa cells, the expression level of the marker gene FOXO1 was exactly the opposite of that in non-pregnant granulosa cells, being high in the early stage and gradually decreasing from the middle to the later stage. Additionally, the expression level of FST in pregnant granulosa cells increased sharply at the later stage. Subsequently, based on the upregulated differentially expressed genes of granulosa cells under different physiological conditions and their expression levels in the pseudotime heatmap, similar trends of representative genes could be observed in the gene expression patterns along the pseudotime axis (Figure 5F).

### 3.7. Differential Analysis and Developmental Trajectory of Ovarian Luteal Cells Under Different Physiological Conditions

In the transcriptomic differential analysis of ovarian luteal cells between pregnant and non-pregnant states, a total of 60 genes were found to be significantly upregulated in the luteal cells of pregnant ovaries, while 52 genes were significantly downregulated (Figure 6A). Further KEGG pathway enrichment analysis of these differentially expressed genes revealed that these genes were widely involved in multiple biological processes, including cortisol synthesis and secretion, ovarian steroidogenesis, steroid hormone biosynthesis, and aldosterone synthesis and secretion. This suggests that these pathways may play crucial roles in the maintenance and regulation of luteal function during pregnancy (Figure 6B). Further pseudotime trajectory analysis of luteal cells under different physiological conditions revealed that there were two major branching nodes in the developmental trajectory of luteal cells during pregnancy, while three branching nodes were observed in the non-pregnant state (Figure 6C). To explore the dynamic transcriptomic changes involved in the differentiation process of luteal cells, we constructed their developmental trajectory. As shown in the figure, the heatmap of representative genes revealed the dynamic expression patterns of these genes along the pseudotime axis, demonstrating their time-dependent and progressive regulatory characteristics at different developmental stages (Figure 6D). The high expression level of genes indicated that they performed specific biological functions during the corresponding pseudotime periods. Specifically, the expression of LHCGR in non-pregnant granulosa cells was downregulated in the early stage and then significantly increased, while its expression trend during pregnancy showed the opposite pattern. Additionally, in pregnant luteal cells, the representative steroid synthesis-related genes HSD3B1 and CYP11A1 had relatively high expression levels in the early stage, followed by a gradual decline and a significant reduction in the late stage. The above results are consistent with the gene expression patterns along the pseudotime axis, demonstrating similar dynamic trends of representative genes along the developmental trajectory under different physiological conditions (Figure 6E).

## 4. Discussion

The remodeling of the mammalian ovary plays a crucial role in the in-depth understanding of physiological processes such as folliculogenesis, ovulation, and corpus luteum formation. Based on this, in this study, we successfully constructed a cell type-specific molecular atlas of the Holstein cow ovary using single-nucleus RNA sequencing technology and accurately identified nine different cell types, including oocytes, granulosa cells, luteal cells, endothelial cells, smooth muscle cells, stromal cells, macrophages, T/NK cells, and epithelial cells (Figure 2A). We systematically compared the differences in cell composition, gene expression characteristics, and intercellular communication networks of bovine ovarian tissues between the pregnant and non-pregnant states, thereby revealing the dynamic regulatory mechanisms of the ovary under these two physiological conditions.

During the process of corpus luteum formation, there is a close relationship between metabolic demand and vascular distribution. As a highly dynamic organ in the mammalian reproductive system, the corpus luteum requires a large amount of cell turnover and energy supply for its rapid growth and development. Therefore, it has a relatively high metabolic rate and extremely high blood supply requirements [47]. The vascular network of the corpus luteum (CL) is crucial for the production of progesterone and the maintenance of pregnancy [48]. Endothelial cells account for up to 61.22% in the non-pregnant state, while dropping to as low as 29.89% during pregnancy, which may reflect the remodeling of the vascular network and the dynamic adjustment of blood flow distribution. This phenomenon may be related to the fact that the non-pregnant ovary needs to maintain a basic vascular network to support follicular development. After the formation of the corpus luteum during pregnancy, the vascular distribution may be more concentrated in the luteal tissue, resulting in a relatively lower proportion of endothelial cells. In addition, the proportion of granulosa cells significantly increases during pregnancy (21.57% vs. 2.17%), which may be associated with the active follicular development in the early stage of pregnancy or the secondary luteinization process. Vascular endothelial growth factor (VEGF) is mainly secreted by granulosa cells and luteal cells during pregnancy, while it is secreted by endothelial cells in the non-pregnant state. This shift may correspond to the angiogenesis requirements at different stages and also reflects the need for multicellular cooperation in corpus luteum maintenance. Regarding the changes in the proportion of immune cells, there are more immune cells in the non-pregnant state, and the number decreases during pregnancy. Considering the local role of immune cells during corpus luteum regression, it is possible that the corpus luteum during pregnancy requires immune privilege, and the reduction of macrophages and other immune cells helps maintain maternal-fetal tolerance [49].

Single-cell transcriptome analysis of human ovarian tissue has clearly revealed its complex cellular composition. It mainly encompasses the following characteristic cell populations: oocytes derived from the germ cell lineage; granulosa cells and theca cells closely associated with follicular development; stromal cells that play a supporting role; endothelial cells and perivascular cells that constitute the vascular system; smooth muscle cells with contractile functions; and various immune cells involved in immune regulation [25,26,50]. Studies have shown that the cell types in the monkey ovary are remarkably similar to those in the human ovary [27,28]. Additionally, in a study of the yak ovary, a total of seven cell types were annotated, namely stromal cells, theca cells, endothelial cells, smooth muscle cells, natural killer cells, proliferating cells, and macrophages [51]. The comprehensive results of the above studies indicate that the cell types of mammalian ovaries are relatively conserved, which to some extent implies the consistency of mammalian ovaries in biological functions.

The mammalian ovary plays a crucial role in the production of sex hormones. Analysis of the differences between ovarian granulosa cells and luteal cells under different physiological states reveals that, compared with the non-pregnant state, genes such as AMH and FST are upregulated in ovarian granulosa cells during pregnancy, and genes such as HSD3B1, LHCGR, and CYP11A1 are upregulated in luteal cells during pregnancy (Figure 5B and Figure 6A). The increased expression of AMH during pregnancy can maintain ovarian reserve, prevent excessive follicular consumption, and reserve resources for postpartum restoration of fertility [51]. Studies have confirmed that the AMH level is positively correlated with ovarian reserve, and its upregulation is a molecular marker of follicular arrest during pregnancy [51,52]. The FST gene (follistatin) mainly acts as a binding protein for members of the TGF-β superfamily in ovarian granulosa cells. By inhibiting the activity of signaling molecules such as activin, it regulates ovarian function. The increase in FST during pregnancy may cooperate with AMH to inhibit follicular development and avoid competition for resources with the embryo. Animal models have shown that FST deficiency leads to premature ovarian failure [53], confirming its key role in maintaining pregnancy homeostasis. Luteinizing hormone receptor (LHCGR) is a G-protein-coupled receptor that primarily functions in the reproductive system and is crucial for the maturation of reproductive organs and embryonic development. In the ovaries of adult women, LHCGR is expressed in theca cells, mature granulosa cells, stromal cells, and luteal cells. When luteinizing hormone (LH) binds to and activates LHCGR, it stimulates theca cells to synthesize and secrete androgens. Subsequently, these androgens serve as substrates and are converted into estradiol by aromatase within granulosa cells. Moreover, the activation of LHCGR is also essential for ovulation and the production of progesterone after corpus luteum formation [54]. The 3β-hydroxysteroid dehydrogenase-1 (3β-HSD1) encoded by the HSD3B1 gene plays a crucial role in the biosynthesis of steroid hormones. This enzyme mainly catalyzes the conversion of Δ5-3β-hydroxysteroid precursors to Δ4-ketosteroids, which is a key step in the synthesis of all active steroid hormones (such as androgens, estrogens, and progesterone) [55]. In addition, HSD3B1 also undertakes important physiological functions in placental tissue and actively participates in the metabolic process of oxidized steroids [56]. Given the crucial position of the placenta in the physiological communication between the mother and the fetus, this function of HSD3B1 may have a potential and profound impact on the normal development of the fetus and the complex interactions between the mother and the fetus.

Luteal cells play a key role in the physiological process of maintaining pregnancy. The differential expression patterns of genes such as CYP11A1 and HSD3B1 in ovarian tissue during pregnancy are key adaptive characteristics of reproductive strategies formed by mammals during long-term evolution. On the one hand, it can precisely ensure the normal development and growth of the current embryo, providing a stable and suitable physiological environment for the embryo. On the other hand, from a long-term perspective, it can maintain the health of the maternal reproductive system and long-term fertility potential, ensuring that the mother still has good fertility after completing the current pregnancy, thus achieving the rational allocation and efficient utilization of reproductive resources [57]. The genes GDF9, FIGLA, and ZP3 also show highly specific expression in the oocytes of humans, monkeys, and mice [25,28,58]. However, the specific expression of these markers was not detected in the oocytes of Holstein cow ovaries. In contrast, CENPF and TOP2A are specifically expressed in the oocytes of Holstein cows, suggesting their crucial roles in maintaining cell cycle stability during the luteal phase and preparing for the development of follicles in the next cycle. These findings are highly consistent with the functional characteristics of the corresponding cells in the mammalian ovary.

In the current situation where the database of bovine ovary-specific marker genes is scarce, this study adopted a method that combines the annotation results of SingleR, the expression characteristics of marker genes, and the relevant literature for cell type annotation. The expression characteristics of specific marker genes for each cell type were visualized using UMAP and violin plots. The results showed that these marker genes exhibited distinct expression patterns, which further verified the reliability of the cell type annotation results. Based on the above reliable cell type annotation, this study successfully depicted a panoramic map of bovine ovarian cells.

However, this study has certain limitations. Due to the limited sample size, it may affect the universality and reliability of the research conclusions. Considering the above limitations, it is necessary to conduct in-depth research with a large sample size in future studies. By increasing the sample size, it is possible to more comprehensively and accurately capture the characteristics and change patterns of ovarian cells, thereby validating and expanding the findings of this study and providing a more solid foundation for research related to bovine ovaries.

## 5. Conclusions

In this study, we successfully constructed a single-cell transcriptomic atlas of the bovine ovary using single-nucleus transcriptome sequencing technology. This achievement offers a novel perspective for the in-depth exploration of the cellular composition and functions of the bovine reproductive system. This study conducted a detailed analysis of the characteristics and dynamic evolution of cell types in the bovine ovary under different physiological states, with a particular emphasis on the differences between the pregnant and non-pregnant states. Through precise cell type annotation and systematic signaling pathway analysis, we determined that, during pregnancy, the numbers of granulosa cells and luteal cells in the ovary increased significantly, while the proportion of endothelial cells decreased. Meanwhile, this study revealed that during pregnancy, oocytes exhibited the enrichment of cell cycle-related pathways. Based on this finding, we further elucidated the mechanism underlying the maintenance of the meiotic arrest state in oocytes. In addition, we constructed an intercellular communication network, which demonstrated at the network level that there are complex and intricate interactions among the identified cell types. This provides important evidence for a comprehensive understanding of the intercellular regulatory mechanisms in the bovine reproductive system.

## Figures and Tables

**Figure 1 genes-16-00931-f001:**
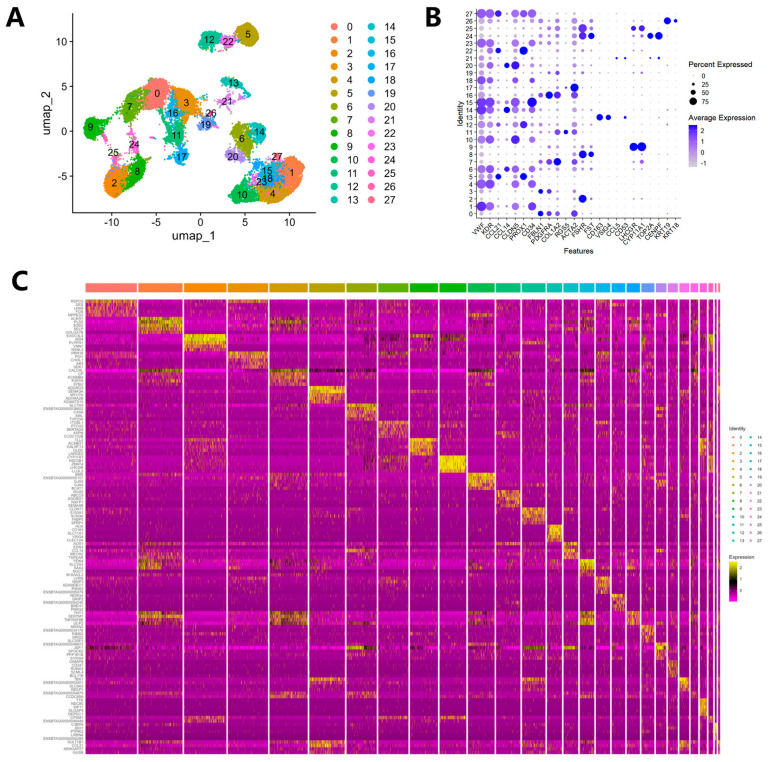
Cell clusters and their characteristic marker genes in the ovaries of Holstein cows. (**A**) Scatter plot from uniform manifold approximation and projection (UMAP) visualizes various cell clusters. Each dot corresponds to a single cell, and the color-coding is based on its cluster membership. (**B**) Dot plot shows the expression characteristics of the selected marker genes in each cell cluster. Gene expression levels from low to high are indicated by a color gradient from light purple to dark blue. The percentage of cells expressing a specific gene is represented by the size of the dots. (**C**) Heatmap shows the differential expression patterns of the top 5 differentially expressed genes (DEGs) in each cell cluster. Gene expression levels from low to high are indicated by a color gradient from purple to yellow.

**Figure 2 genes-16-00931-f002:**
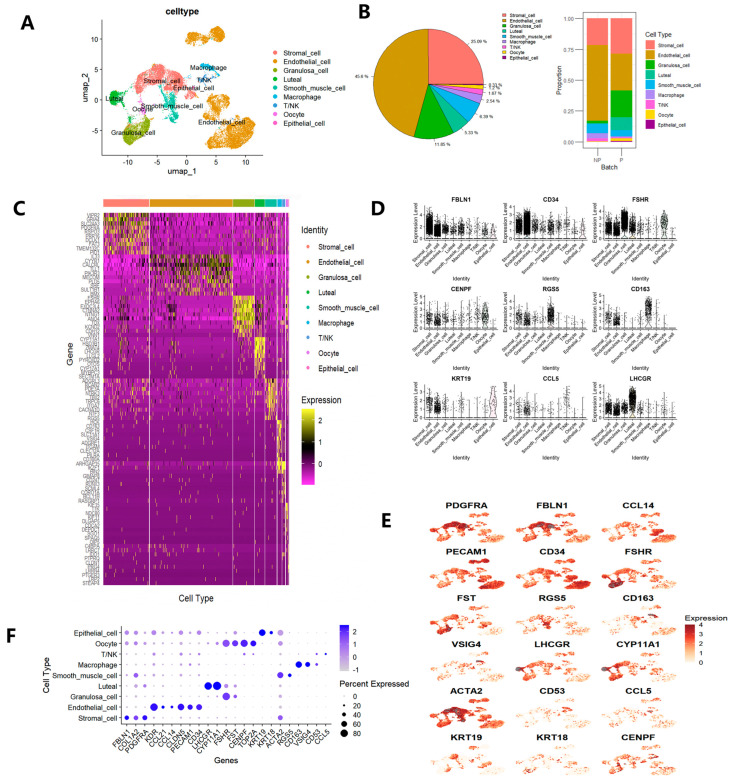
Cell types and their characteristic marker genes in the ovaries of Holstein cows. (**A**) Uniform Manifold Approximation and Projection (UMAP) scatter plot visualizing various cell types. Each dot corresponds to a single cell, which is color-coded according to its cell type membership. (**B**) A pie chart shows the proportion of different cell types, and a bar chart shows the proportions of cell types under different physiological conditions. (**C**) Heatmap displays the differential expression patterns of the top 10 DEGs in each cell type. Gene expression levels from low to high are indicated by a color gradient from purple to yellow. (**D**) Violin plots show the expression specificity of the characteristic genes for each cell type. The expression values of the marker genes are scaled by logarithmic normalization. The vertical axis shows the expression scores of the marker genes. (**E**) Feature plots show the expression specificity of the characteristic genes across all ovary cells. The expression levels of each gene, ranging from none to high, are represented by a color gradient from light beige to dark red. (**F**) Dot plot shows the differential expression patterns of the selected characteristic genes for each cell type. Gene expression levels from low to high are indicated by a color gradient from light purple to dark blue. The percentage of cells expressing a specific gene is represented by the size of the dots.

**Figure 3 genes-16-00931-f003:**
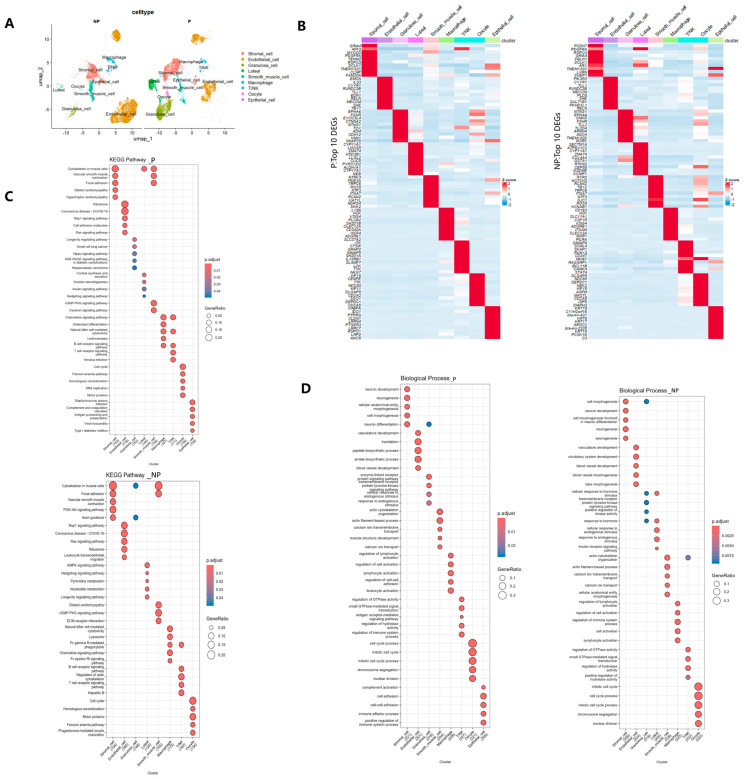
Comparison of expression profiles of ovarian cell types in Holstein cows under different physiological conditions. (**A**) Differences in somatic cell types visualized by UMAP. (**B**) The top 10 differentially expressed genes (DEGs) for each ovarian somatic cell type under different physiological conditions. The gene expression levels are represented by a color gradient from blue (low expression) to red (high expression). (**C**) A bubble plot showing the top five enriched pathways for each cell type in the Kyoto Encyclopedia of Genes and Genomes (KEGG) analysis. The proportion of genes involved in the corresponding pathway is represented by the size of the bubble. The adjusted *p*-values from low to high are represented by a color gradient from blue to red. (**D**) A bubble plot showing the top five enriched biological process (BP) terms for each cell type in the gene ontology (GO) analysis. The proportion of genes involved in the corresponding biological process is represented by the size of the bubble. The adjusted *p*-values from low to high are represented by a color gradient from blue to red.

**Figure 4 genes-16-00931-f004:**
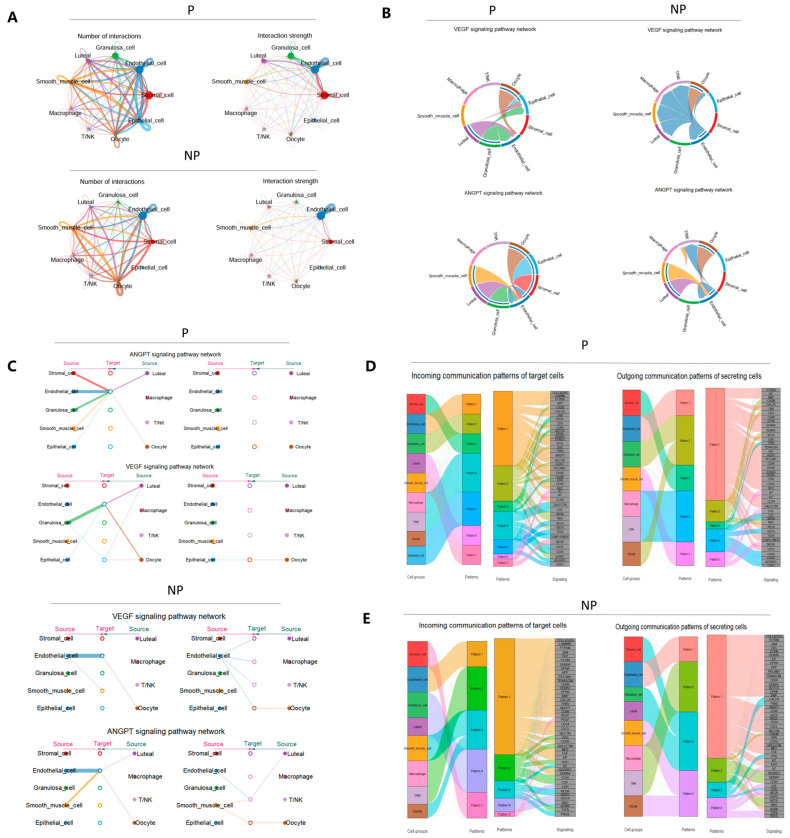
Cell communication analysis reveals signal crosstalk among various cell types in the ovaries of Holstein cows under different physiological states. (**A**) Circle plot representing cell communication among cell types. Circle sizes represent the number of cells, whereas edge widths represent the communication probability. (**B**) Interactions of the VEGF and ANGPT pathways among the main cell populations in the ovaries of Holstein cows under different physiological states. (**C**) Hierarchical plot showing the intercellular communication network for VEGF and ANGPT signaling pathways. Circle sizes represent the number of cells, whereas edge widths represent the communication probability. (**D**) Putative incoming and outgoing communication patterns of secretory cells in the bovine ovary under the pregnant physiological state. (**E**) Putative incoming and outgoing communication patterns of secretory cells in the bovine ovary under the non-pregnant physiological state.

**Figure 5 genes-16-00931-f005:**
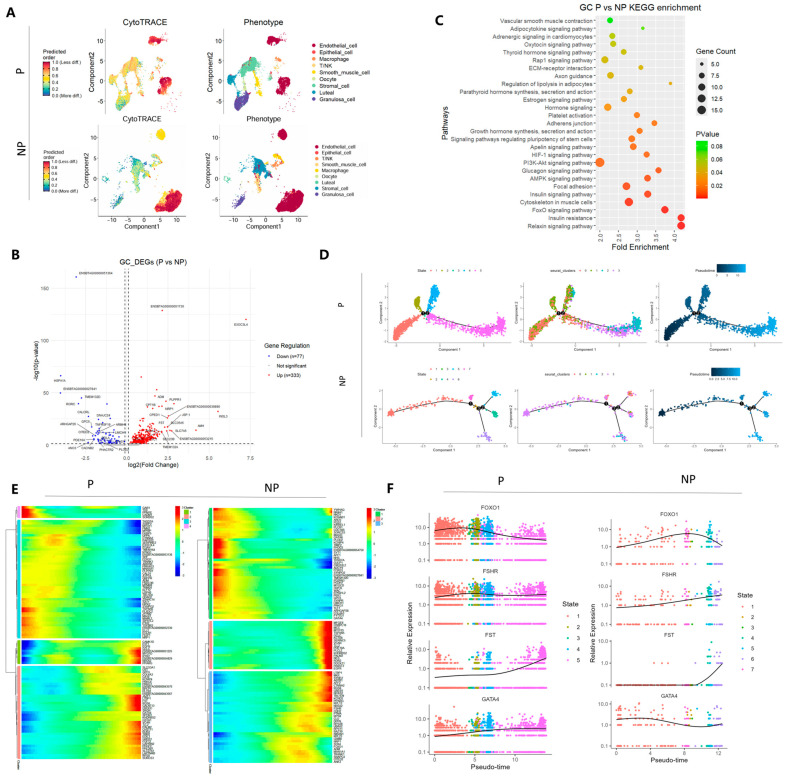
Analysis of ovarian cell differentiation, differential analysis of granulosa cells, and pseudotime analysis during pregnancy and non-pregnancy. (**A**) Analysis of the differentiation potential of ovarian cells using CytoTRACE. The gradient color from red to blue indicates the differentiation ability from low to high. (**B**) Volcano plot of differential gene analysis of ovarian granulosa cells between pregnant and non-pregnant Holstein cows. Red dots represent upregulated differentially expressed genes (DEGs), and blue dots represent downregulated DEGs. (**C**) KEGG enrichment analysis showing the pathways enriched by differentially expressed genes in granulosa cells (GCs). (**D**) Pseudotime trajectory analysis of granulosa cells under different physiological conditions. (**E**) Pseudotime heatmap showing the dynamic gene expression profiles during the fate determination process of GCs. The expression levels of dynamic genes are represented by a color gradient from red (high) to blue (low). (**F**) Expression trend plots of characteristic genes arranged along the pseudotime in granulosa cell subtypes.

**Figure 6 genes-16-00931-f006:**
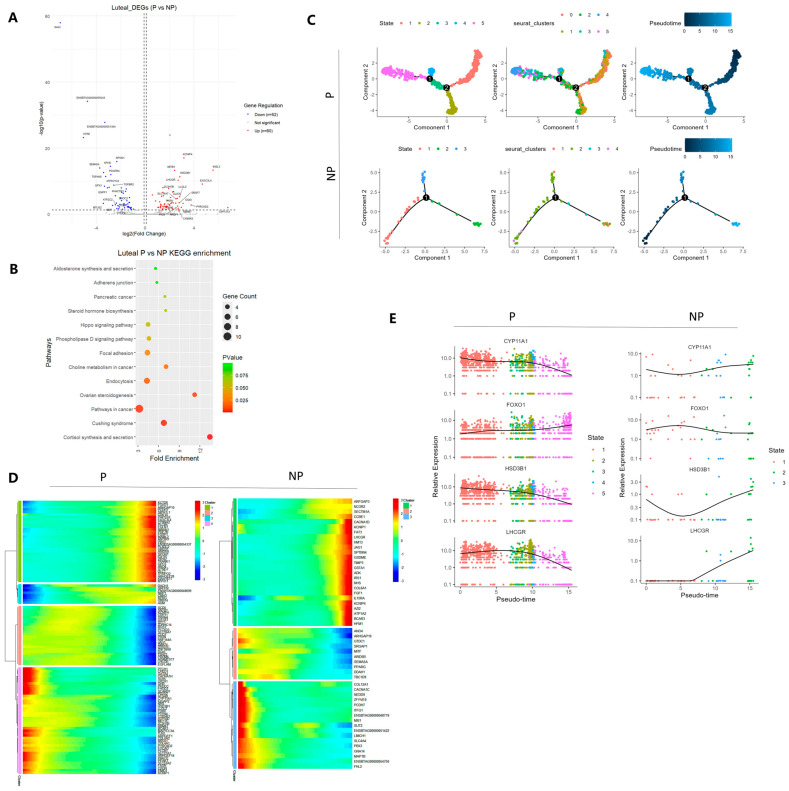
Differential analysis and pseudotime analysis of luteal cells under different physiological conditions. (**A**) Volcano plot of differential gene analysis of ovarian luteal cells between pregnant and non-pregnant Holstein cows. Red represents upregulated differentially expressed genes (DEGs), and blue represents downregulated DEGs. (**B**) KEGG enrichment analysis showing the pathways enriched by the differentially expressed genes in luteal cells. (**C**) Pseudotime trajectory analysis of luteal cells under different physiological conditions. (**D**) Pseudotime heatmap shows the dynamic gene expression profiles during the fate determination process of luteal cells. The expression levels of dynamic genes are represented by a color gradient from red (high) to blue (low). (**E**) Expression trend plots of characteristic genes arranged along the pseudotime in luteal cell subtypes.

## Data Availability

The data from this study are not publicly available. As the data are still under further analysis, they are not provided to the public for the time being. Subject to relevant regulations and obtaining the necessary authorization, data sharing may be considered in the future. For specific data requests, please contact the corresponding author for further communication.

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
