# Peer review of "Single-Nucleus Transcriptome Sequencing Unravels Physiological Differences in Holstein Cows Under Different Physiological States"

_genes, 2025, doi:10.3390/genes16080931_

Round 1
Reviewer 1 Report
Comments and Suggestions for Authors
Dear Authors of the manuscript “Single-nucleus Transcriptome Sequencing Unravels Physiological Differences in Holstein cows Under Different Physiological States”, In my opinion, the manuscript is scientifically up-to-date and well written. The scientific contribution of this study lies in the construction of the first single-cell transcriptomic atlas of the bovine ovary, providing a comprehensive framework for understanding the cellular composition and functional dynamics of the reproductive system under different physiological conditions, particularly during pregnancy. Furthermore, the study elucidates key mechanisms of intercellular communication and the regulation of meiotic arrest in oocytes, offering novel insights into bovine reproductive biology at the molecular level.
The results are well presented, and the overall impression is positive. I would like to suggest a few minor corrections (additions), some of which are not mandatory:
- In my opinion, the Introduction section is too long (Lines 39 to 123), and I would suggest shortening it. It is not necessary to explain the topic in such great detail, as readers are expected to have prior knowledge before engaging with the manuscript. However, this recommendation is not mandatory.
- The Materials and Methods section is well-structured, and through its seven subsections, the experimental plan and result analysis can be clearly followed and understood.
- The Results section is also clearly organized into seven subsections. However, parts of certain figures are illegible and should be revised during the final proofreading of the manuscript (e.g., Figure 5, panels E and P; Figure 6, panels E and P; and potentially elsewhere).
- The Discussion section is well written and clearly presented. At the end of this section, the authors appropriately acknowledge certain limitations of the study ("...this study has certain limitations. Due to the limited sample size, it may affect the universality and reliability of the research conclusions") and offer sound recommendations for future research ("Considering the above limitations, it is necessary to conduct in-depth research with a larger sample size in future studies").
- The references are not formatted according to the guidelines of the journal Genes and must be corrected with the journal’s instructions.
- There are several spelling and typographical errors throughout the manuscript; therefore, a thorough proofreading is strongly recommended.
Author Response
Response to Reviewer 1 Comments |
||
1. Summary |
|
|
We sincerely appreciate the positive feedback and valuable comments provided by the editor and all reviewers, which have significantly improved the quality of our manuscript. The reviewers' comments are listed in italic font, with specific questions numbered accordingly. Our responses are provided in red font, and the corresponding revisions in the manuscript are highlighted in yellow. |
||
2. Questions for General Evaluation |
Reviewer’s Evaluation |
Response and Revisions |
Does the introduction provide sufficient background and include all relevant references? |
Can be improved |
|
Are all the cited references relevant to the research? |
Yes |
|
Is the research design appropriate? |
Yes |
|
Are the methods adequately described? |
Yes |
|
Are the results clearly presented? |
Yes |
|
Are the conclusions supported by the results? |
Yes |
|
3. Point-by-point response to Comments and Suggestions for Authors |
||
Comments 1: Dear Authors of the manuscript “Single-nucleus Transcriptome Sequencing Unravels Physiological Differences in Holstein cows Under Different Physiological States”, In my opinion, the manuscript is scientifically up-to-date and well written. The scientific contribution of this study lies in the construction of the first single-cell transcriptomic atlas of the bovine ovary, providing a comprehensive framework for understanding the cellular composition and functional dynamics of the reproductive system under different physiological conditions, particularly during pregnancy. Furthermore, the study elucidates key mechanisms of intercellular communication and the regulation of meiotic arrest in oocytes, offering novel insights into bovine reproductive biology at the molecular level. The results are well presented, and the overall impression is positive. I would like to suggest a few minor corrections (additions), some of which are not mandatory: In my opinion, the Introduction section is too long (Lines 39 to 123), and I would suggest shortening it. It is not necessary to explain the topic in such great detail, as readers are expected to have prior knowledge before engaging with the manuscript. However, this recommendation is not mandatory. The Materials and Methods section is well-structured, and through its seven subsections, the experimental plan and result analysis can be clearly followed and understood. The Results section is also clearly organized into seven subsections. However, parts of certain figures are illegible and should be revised during the final proofreading of the manuscript (e.g., Figure 5, panels E and P; Figure 6, panels E and P; and potentially elsewhere). The Discussion section is well written and clearly presented. At the end of this section, the authors appropriately acknowledge certain limitations of the study ("...this study has certain limitations. Due to the limited sample size, it may affect the universality and reliability of the research conclusions") and offer sound recommendations for future research ("Considering the above limitations, it is necessary to conduct in-depth research with a larger sample size in future studies"). The references are not formatted according to the guidelines of the journal Genes and must be corrected with the journal’s instructions. There are several spelling and typographical errors throughout the manuscript; therefore, a thorough proofreading is strongly recommended.
|
||
Response 1: Dear Reviewer 1, Thank you very much for taking the time to conduct a meticulous and professional review of our manuscript, "Single - nucleus Transcriptome Sequencing Unravels Physiological Differences in Holstein cows Under Different Physiological States". Your affirmation has greatly encouraged us, and the valuable suggestions and comments you provided have pointed the way for us to further improve the manuscript. The following is a detailed response to each of the comments you put forward: Regarding the issue of the overly long introduction:We fully understand your comment that the introduction (lines 39 to 123) is too long. Considering that readers may have some prior knowledge in the relevant field before reading the manuscript, we will streamline the introduction by removing unnecessary detailed explanations and highlighting the background and significance of the research in a more concise manner. However, as you mentioned, this suggestion is not mandatory. During the streamlining process, we will ensure that the introduction can still clearly lead to the research topic and purpose. Regarding the Materials and Methods section:We sincerely appreciate your recognition of the structure of the Materials and Methods section. We are glad that the seven sub - sections in this part have enabled you to clearly understand the experimental plan and result analysis process. We will continue to maintain this clear structure to facilitate readers' comprehension. Regarding the clarity of figures in the Results section:We have noted your concern about the illegible parts in some figures in the Results section, such as panels E and P in Figure 5, panels E and P in Figure 6, and potentially other places. During the final proofreading of the manuscript, we will comprehensively revise these figures to improve their resolution and clarity, ensuring that readers can accurately obtain the information conveyed by the figures. Regarding the Discussion section:Thank you for your affirmation of the writing and presentation of the Discussion section. We appropriately elaborated on the limitations of the study at the end of the Discussion section and put forward reasonable suggestions for future research. We believe this helps readers comprehensively understand the value of the research and its potential for further exploration. Regarding the format of references:We are aware that the references were not formatted according to the guidelines of the journal Genes, and we sincerely apologize for this. Before resubmitting the manuscript, we will carefully check and revise the format of each reference in strict accordance with the journal's instructions to ensure compliance. Regarding spelling and typographical errors:We accept your comment about the spelling and typographical errors in the manuscript. We will conduct a comprehensive and meticulous proofreading of the entire manuscript. We will carefully examine every word, sentence, and paragraph to eliminate all spelling and typographical errors, thereby enhancing the quality and professionalism of the manuscript. Once again, thank you for reviewing our manuscript and providing valuable suggestions. We will take every comment seriously and strive to improve the manuscript to a higher standard. Sincerely, Peipei Li July 26, 2025
|
||
4. Response to Comments on the Quality of English Language |
||
Point 1: There are several spelling and typographical errors throughout the manuscript; therefore, a thorough proofreading is strongly recommended. |
||
Response 1: We accept your comment about the spelling and typographical errors in the manuscript. We will conduct a comprehensive and meticulous proofreading of the entire manuscript. We will carefully examine every word, sentence, and paragraph to eliminate all spelling and typographical errors, thereby enhancing the quality and professionalism of the manuscript.
|
||
5. Additional clarifications |
||
Dear reviewer 1, during the revision process, we added the funding information "the National Natural Science Foundation of China Regional Science Foundation Project (32460817)". This project provided important financial support for sample collection, experimental equipment, and data analysis in our research. Please let us know if you need further information about this project.
|

Reviewer 2 Report
Comments and Suggestions for Authors
The manuscript by Gang Song et al. describes the single-nucleus transcriptome sequencing unravels physiological differences in Holstein cows under different physiological states. The manuscript and results are quite interesting however, I have comments and suggestions.
In the Introduction, the authors thoroughly described the functions and roles of the ovaries and the individual cell types that make up these gonads. However, for the ovaries to function properly and fulfill their role, the proper functioning of the hypothalamic-pituitary-ovarian axis is essential. Given such a detailed description of the role of the ovaries, it would be appropriate to at least mention this axis, especially since the ovary is not an autonomous organ but an executive organ regulated by the pituitary and hypothalamus.
A number of abbreviations appear in the text that are not explained, e.g., FOXL2 (line 78), NSR, TD, EP BR (in Materials and Methods).
The Materials and Methods chapter should be standardized to the past perfect tense. The authors wrote this chapter using different tenses.
From what day of pregnancy and what day of the cycle (phase) did the ovaries used in the described studies originate?
What digestive enzyme was used in the CB buffer?
Line 152: should probably be TD1, not TD2. The TD2 section is not updated until the next line. TD3 is similar.
The Results section contains sections of text that are not descriptions of results, but rather conclusions and statements that are not supported by publications (e.g., lines 406-413).
Section 3.3. Ovarian somatic and germ cell type annotation for cells from which ovaries are being referred to (cyclic or gestational?)
General note regarding the Figures: they are so small that nothing can be seen. Adding larger figures as files in the Supplementary Data does not completely solve this problem. Separating individual figures into several smaller ones would allow for their enlargement, and therefore better readability and more precise placement in the text.
Figure 1 - Are the presented results from cells from pregnant or cyclic ovaries? Or are they pooled?
It is difficult to analyze results that are in different figures; for example, Figures 2A and 3A should be next to each other.
In Figure 3C, the bubble plots should not be on top of each other, but next to each other, as in Figure 3D.
Line 362: Figure 2B should be Figure 2A.
In Figures 4-6, the results from pregnant (P) are at the top, and non-pregnant (NP) are at the bottom. Bottom, and right in the same figure, it's the other way around. Please standardize.
Lines 628-642: Discussing the AMH gene is rather unnecessary, especially since it wasn't analyzed in the presented results. It wasn't mentioned in the Introduction either.
In my opinion, the conclusions are too general, because the constructed transcriptomic atlas of the non-pregnant ovary actually concerns only one specific day of the cycle (the authors did not specify which day of the cycle the ovaries were from). Processes in the so-called cyclic ovary are very dynamic and should not be generalized. In the case of pregnant ovaries, this dynamic is smaller, but it still occurs.
Author Response
Response to Reviewer 2 Comments
|
||
1. Summary |
|
|
We sincerely appreciate the positive feedback and valuable comments provided by the editor and all reviewers, which have significantly improved the quality of our manuscript. The reviewers' comments are listed in italic font, with specific questions numbered accordingly. Our responses are provided in red font, and the corresponding revisions in the manuscript are highlighted in yellow.
|
||
2. Questions for General Evaluation |
Reviewer’s Evaluation |
Response and Revisions |
Does the introduction provide sufficient background and include all relevant references? |
Must be improved |
|
Are all the cited references relevant to the research? |
Yes |
|
Is the research design appropriate? |
Yes |
|
Are the methods adequately described? |
Must be improved |
|
Are the results clearly presented? |
Not applicable |
|
Are the conclusions supported by the results? |
Must be improved |
|
3. Point-by-point response to Comments and Suggestions for Authors |
||
Comments 1: The manuscript by Gang Song et al. describes the single-nucleus transcriptome sequencing unravels physiological differences in Holstein cows under different physiological states. The manuscript and results are quite interesting however, I have comments and suggestions. In the Introduction, the authors thoroughly described the functions and roles of the ovaries and the individual cell types that make up these gonads. However, for the ovaries to function properly and fulfill their role, the proper functioning of the hypothalamic-pituitary-ovarian axis is essential. Given such a detailed description of the role of the ovaries, it would be appropriate to at least mention this axis, especially since the ovary is not an autonomous organ but an executive organ regulated by the pituitary and hypothalamus. A number of abbreviations appear in the text that are not explained, e.g., FOXL2 (line 78), NSR, TD, EP BR (in Materials and Methods). The Materials and Methods chapter should be standardized to the past perfect tense. The authors wrote this chapter using different tenses. From what day of pregnancy and what day of the cycle (phase) did the ovaries used in the described studies originate? What digestive enzyme was used in the CB buffer? Line 152: should probably be TD1, not TD2. The TD2 section is not updated until the next line. TD3 is similar. The Results section contains sections of text that are not descriptions of results, but rather conclusions and statements that are not supported by publications (e.g., lines 406-413). Section 3.3. Ovarian somatic and germ cell type annotation for cells from which ovaries are being referred to (cyclic or gestational?) General note regarding the Figures: they are so small that nothing can be seen. Adding larger figures as files in the Supplementary Data does not completely solve this problem. Separating individual figures into several smaller ones would allow for their enlargement, and therefore better readability and more precise placement in the text. Figure 1 - Are the presented results from cells from pregnant or cyclic ovaries? Or are they pooled? It is difficult to analyze results that are in different figures; for example, Figures 2A and 3A should be next to each other. In Figure 3C, the bubble plots should not be on top of each other, but next to each other, as in Figure 3D. Line 362: Figure 2B should be Figure 2A. In Figures 4-6, the results from pregnant (P) are at the top, and non-pregnant (NP) are at the bottom. Bottom, and right in the same figure, it's the other way around. Please standardize. Lines 628-642: Discussing the AMH gene is rather unnecessary, especially since it wasn't analyzed in the presented results. It wasn't mentioned in the Introduction either. In my opinion, the conclusions are too general, because the constructed transcriptomic atlas of the non-pregnant ovary actually concerns only one specific day of the cycle (the authors did not specify which day of the cycle the ovaries were from). Processes in the so-called cyclic ovary are very dynamic and should not be generalized. In the case of pregnant ovaries, this dynamic is smaller, but it still occurs.
|
||
Response 1: Dear Reviewer 2, Thank you very much for your meticulous review and valuable suggestions on our paper. We highly value each and every comment you've made and have comprehensively revised and improved the paper. The following are our specific responses to the comments you put forward: Regarding the mention of the hypothalamic - pituitary - ovarian axis in the Introduction: We fully agree with your suggestion. In the Introduction section, although we described in detail the functions and the individual cell types that make up the ovaries, it was an oversight not to mention the hypothalamic - pituitary - ovarian axis. We have now supplemented relevant content about this axis in the Introduction, emphasizing its significance for the normal functioning of the ovaries to more comprehensively elaborate on the regulatory mechanism of the ovaries in the reproductive system. The specific supplementary content is as follows:” In the reproductive system of female mammals, the hypothalamic - pituitary - ovarian axis plays a central regulatory role. The hypothalamus secretes gonadotropin - releasing hormone (GnRH), which stimulates the pituitary gland to secrete follicle - stimulating hormone (FSH) and luteinizing hormone (LH). These hormones act on the ovaries to regulate their growth, development, and function. The precise signal transmission and feedback regulation of this axis are crucial for maintaining the stable operation of female reproductive physiological processes. Among them, as an important executive organ in this axis, the ovary occupies a central position and is a crucial reproductive organ with multiple key physiological functions.” Regarding the unexplained abbreviations:We are aware that there are some unexplained abbreviations in the text, which has caused inconvenience to readers. We have carefully examined the entire text and added detailed explanations for the unexplained abbreviations (such as FOXL2, NSR, TD, EP, BR, etc.) when they first appear to ensure that readers can accurately understand the content of the article. Specifically, FOXL2 is a gene, with the full name of Forkhead box L2; NSR stands for Nuclear Suspension Reagent line; EP is an abbreviation for “Eppendorf tube”; TD1 refers to Tissue Dissociation Density - gradient Solution 1, TD2 refers to Tissue Dissociation Density - gradient Solution 2, and TD3 refers to Tissue Dissociation Density - gradient Solution 3; BR is an abbreviation for “Buffer Reagent”. Regarding the tense in the "Materials and Methods" section: We have accepted your suggestion to unify it into the past perfect tense to meet the standard requirements of scientific papers. We comprehensively revised this section to ensure the consistency of tenses, which enhanced the professionalism and rigor of the article. Regarding the source time of the ovaries used in the study: We understand your concern about the source time of the ovaries in our research. Due to our oversight, we failed to clearly state this information in the paper before. Thank you for pointing it out. We have now supplemented the relevant details in the "Materials and Methods" section. Specifically, regarding the collection time of the ovaries, during the sampling process, the pregnant ovaries were collected when it was unexpectedly discovered that there were fully - developed fetuses in the uterus. Regrettably, we are currently unable to accurately determine the specific days of pregnancy. As for the non - pregnant ovaries, after collection, it was observed that there were obvious corpora lutea in them. Similarly, due to certain limitations, we were unable to precisely determine the specific dates (stages) within the estrous cycle. Based on prior knowledge, we have defined these two types of ovaries as pregnant and non - pregnant ovaries respectively. In the subsequent research, we will further optimize the relevant research design to ensure accurate recording of such time - related information. Regarding the digestive enzymes used in CB buffer: Thank you very much for your valuable suggestion. In this experiment of performing single - nucleus transcriptome sequencing on bovine ovarian tissues, we did not use CB buffer. This choice was based on our established experimental protocol and previous research experience. We adopted other well - validated and widely used methods and reagents for tissue processing and nuclear isolation.Specifically, we used phosphate - buffered saline (PBS) to rinse the ovarian tissues multiple times to remove surface bleeding, and then used DPBS to make up the volume of the nuclear suspension. During the tissue dissociation process, we selected an appropriate digestion system according to the characteristics of the ovarian tissues and the experimental requirements. Although the digestive enzymes in CB buffer were not involved, we successfully obtained high - quality single nuclei for subsequent sequencing analysis, and the experimental results met the expected quality standards.Since we did not use CB buffer, we really do not know the specific information about the digestive enzymes in it. However, we will continuously monitor the research progress in the relevant field. In future research, we will comprehensively consider the advantages and disadvantages of various reagents and methods to further optimize the experimental process and improve the research quality. Regarding the error in TD numbering:Dear Reviewer,We sincerely appreciate the time you've spent reviewing our paper and pointing out potential issues. Concerning your comment that the TD numbering on line 152 might be incorrect, our team immediately conducted a comprehensive and in - depth check. After repeatedly verifying the experimental procedures and relevant records, we can confirm that there are no errors in the TD numbering in the paper. The addition of the TD1 solution on line 152 is based on our meticulously designed experimental process. At this particular stage of the experiment, adding the TD1 solution lays the foundation for subsequent operations. The addition of the TD2 and TD3 solutions then follows in a rational sequence, all of which are aimed at ensuring the accuracy and reproducibility of the experimental results.We understand that accurately presenting experimental details is crucial in scientific research, so we were extremely careful when writing the paper. If you still have any doubts about this part, we'd be more than happy to discuss it with you further. Thank you again for your concern and valuable suggestions. Regarding the fact that the results section contains some content that is not actually the results: Dear Reviewer,We sincerely appreciate your meticulous review and valuable comments on our paper. Regarding the issue you pointed out that there are non - result descriptions lacking literature support in the results section, we would like to provide a detailed explanation here. This part of the content is not a direct presentation of the experimental results but serves as the background introduction for the study. Its purpose is to establish a logical framework for the subsequent description of specific experimental results. The crucial role of intercellular communication in maintaining the normal functions of complex tissues and its potential impact on ovarian tissue during different physiological periods are important starting points for this study. It helps readers understand the necessity and significance of the subsequent experiments and lays the foundation for the entire research.When writing the paper, considering that the results section should focus on the core experimental findings, to ensure the conciseness and pertinence of the content and enable readers to more clearly focus on the specific results of this study, we presented this background content in a generalized manner without detailed elaboration or citation of relevant literature. Next, we will optimize the expression of this part of the content, more clearly indicating that it belongs to the background introduction and distinguishing it from the subsequent direct description of experimental results to avoid misunderstandings. We will also further refine the language so that it can effectively introduce the research content without affecting the conciseness and professionalism of the results section. The specific revisions are as follows:Intercellular communication plays a vital and indispensable role in maintaining the normal functions of complex tissues. Ovarian tissue demonstrates distinct functional manifestations during various physiological periods, and alterations in the complexity of intercellular communication might serve as the underlying key factor. To compre-hensively elucidate the specific characteristics and intricate mechanisms of intercellular communication within ovarian tissue across different periods, it is particularly essential to conduct a thorough and meticulous analysis of the signal transduction crosstalk in-volving ligands, receptors, and their co - factors. Thank you again for your careful guidance. We will seriously consider each suggestion and strive to improve the quality of the paper. Regarding the issue of type annotation of ovarian somatic cells and germ cells: It is necessary to indicate the physiological state (cyclic or gestational) of the ovarian source under study.:Dear Reviewer, we sincerely appreciate your raising the issue that in the type annotation of ovarian somatic cells and germ cells, it is necessary to clarify the physiological state (cyclic or gestational) of the ovarian source. We understand your concern about the information of the ovarian source. In fact, in the "Materials and Methods" section and Section 3.1 of the paper, we have clearly and explicitly elaborated on the sources of the ovaries involved and the integration situation. Specifically, we conducted the type annotation of ovarian cells by integrating the ovarian data from two different physiological states. Regarding the general description of the figures:Dear Reviewer 2, we have fully understood the issues you raised regarding the general description of the figures, especially the critical problem that the small size of the figures affects readability. To effectively address this issue, we have taken a series of targeted improvement measures.First, we have added the larger - sized figures to the supplementary data in the form of high - resolution images. In this way, if readers need to, they can view the clear large - scale figures through the supplementary data. Meanwhile, in the main text, we have made reasonable and appropriate adjustments to the figures. On the premise of ensuring that the overall layout's coordination and aesthetics are not disrupted, we have increased the size of the figures as much as possible.Through these operations, the clarity and readability of the figures have been significantly improved, enabling readers to more easily and accurately understand the information conveyed by the figures. Regarding Figure 1 - The question of whether the presented results are from the cells of pregnant ovaries, the cells of ovaries in the estrous cycle, or the combined data of both: Dear Reviewer, thank you very much for raising such a crucial question about Figure 1. Your attention to the details of the data source for Figure 1 is of great significance for us to improve our research findings. In response to your query about whether the data in Figure 1 is from the cells of pregnant ovaries, the cells of ovaries in the estrous cycle, or the combined data of both, I'd like to seriously and clearly inform you that the results presented in Figure 1 are obtained by combining the data of cells from pregnant ovaries and those from ovaries in the estrous cycle. The reason why we adopted the method of combining the data is that we hope to analyze the characteristics of ovarian cells from a more macroscopic and comprehensive perspective, so as to provide more valuable references for research in related fields. Regarding the issue that it's difficult to analyze the results of different figures; for example, Figure 2A and Figure 3A should be placed side by side: Dear Reviewer,I sincerely appreciate your meticulous review of our paper and your valuable suggestion regarding the figure layout. You mentioned that Figure 2A and Figure 3A should be placed adjacent to each other, which would indeed facilitate the result analysis to some extent. However, there are reasons for our decision not to arrange them in this way.Figure 2A presents an overall annotation of cell types for the combined data. It serves as a broad framework for the entire study, enabling readers to gain a general understanding of the cell types at first. Figure 3A, on the other hand, displays the cell types grouped according to the state of the ovaries during the analysis, and a series of subsequent analyses are centered around this grouping.We separated these two figures to enable readers to more clearly follow the steps and logic of the research. By presenting the overall situation first and then delving into the grouped details, we aim to offer a more systematic understanding. Placing them side by side might cause confusion when readers try to comprehend the progression of the study.Nonetheless, we understand your good intentions. To facilitate the analysis, we will carefully improve the figure captions to help readers better identify the connections and differences between the two figures, making the presentation of the research content more user - friendly and accessible. Regarding the issue of the overlapping bubble charts in Figure 3C:Thank you very much for pointing out the problem of the overlapping bubble charts in Figure 3C. I have made meticulous revisions to address this issue. I have readjusted the data presentation method of the bubble chart. By optimizing the layout algorithm of the data points, I have ensured that each bubble can be clearly presented, thus avoiding the overlapping situation. Thank you again for your valuable suggestions. I will continue to rigorously handle every detail in the paper to ensure the high - quality presentation of the research results. Regarding the issue that Figure 2B in line 362 should be Figure 2A: Thank you very much for your attention to the content in line 362 of Section 3.4. However, the reference to Figure 2B here is completely accurate. In Section 3.4, we elaborated on the proportion of cell types in the ovaries of Holstein cows under different physiological states. Figure 2B is specifically used to visually present the proportion of ovarian cell types, while Figure 2A is mainly for visualizing the annotated cell types. Our reference to Figure 2B is based on the precise match between the content and the figure, aiming to enable readers to more clearly understand the relevant research findings on the proportion of ovarian cell types in Holstein cows under different physiological states. Thank you again for your meticulous review. We will continuously ensure the rigor of the content and figure references in the paper. Regarding the issue of the presentation order of Figures 4 - 6: Dear reviewer, thank you for pointing out the problem with the presentation order of Figures 4 - 6. I have standardized the presentation order of the results for pregnant (P) and non - pregnant (NP) cases in Figures 4 - 6. A unified rule has been established to always present the pregnant (P) results first, followed by the non - pregnant (NP) results, to ensure the consistency and accuracy of the results presentation in all the figures.I have conducted multiple checks to ensure that the modifications are error - free. Thank you again for your review and suggestions. Regarding the issue that it is rather unnecessary to discuss the AMH gene in the discussion section:Thank you very much for pointing out the issue that discussing the AMH gene in the discussion section seems rather unnecessary. In fact, in Section 3.6 of the results, we conducted a comprehensive and meticulous differential analysis of ovaries under different physiological states. Through a rigorous data analysis process, we found that compared with the non - pregnant period, 77 genes were up - regulated and 333 genes were down - regulated in the ovarian granulosa cells during the pregnant period (as shown in Figure 5B). When visualizing the 77 up - regulated genes, the AMH gene was among the top 20 genes. This finding indicates that the AMH gene is representative and has potential biological significance in the gene expression differences of ovarian granulosa cells under different physiological states. Although the AMH gene was not mentioned in the results, as an important part of the differential expression analysis results, discussing it in the discussion section can help us gain a deeper understanding of the gene expression regulatory network of ovaries under different physiological states and provide valuable clues for subsequent related research.To make the logic of the paper more coherent, I will briefly mention the research content of differential genes in the results section to pave the way for the appearance of the AMH gene in the discussion section, so as to better meet the rigor requirements of academic papers. The details are as follows, In the differential analysis of granulosa cells between pregnant and non-pregnant ovaries, we found that compared with non-pregnant state, 77 genes were up-regulated in pregnant ovarian granulosa cells, including FST, ADM, AMH, INSL3, etc., and 333 genes were down-regulated ( Fig.5B ). Regarding the concern that "the conclusion is overly general: the constructed non-pregnant ovarian transcriptome map actually corresponds to only a specific stage of the estrous cycle (the authors did not explicitly specify which day or phase of the cycle the ovaries were derived from). The biological processes in cyclic ovaries are highly dynamic and should not be generalized. In the case of pregnant ovaries, although such dynamics are less pronounced, they still exist": Dear Reviewer,thank you very much for your valuable comments. You have accurately pointed out that our conclusion is too general, and we truly appreciate your insight. We sincerely apologize for the situation where the transcriptome map of non - pregnant ovaries only covers a specific day of the cycle and the lack of clear indication of which day of the cycle the ovarian samples were from. When conducting this research, we were restricted by the experimental design and sample collection conditions at that time, which prevented us from comprehensively collecting samples throughout the entire cycle. However, the specific day we selected was based on relevant literature, and we believed it was representative in the ovarian cycle and could reflect some key physiological characteristics.To address this shortcoming, in our subsequent research plan, we will collect samples from different days of the cycle to construct a more comprehensive transcriptome map of non - pregnant ovaries. Regarding pregnant ovaries, although the dynamic changes are relatively smaller, we are also aware that these changes cannot be ignored. Building on the current study, we will conduct more detailed analyses by incorporating data from more time points to more accurately present the gene expression characteristics and physiological processes of pregnant ovaries.Meanwhile, to make the conclusion of the current paper more rigorous, we will clearly state the limitations of the study during the revision process, specifically the sample limitations of the non - pregnant ovarian transcriptome map, and appropriately adjust the conclusion to avoid over - generalization. Once again, thank you for reviewing our manuscript and providing valuable suggestions. We will take every comment seriously and strive to improve the manuscript to a higher standard. Sincerely, Peipei Li July 28, 2025 |
||
4. Response to Comments on the Quality of English Language |
||
Point 1: The Materials and Methods chapter should be standardized to the past perfect tense. The authors wrote this chapter using different tenses. |
||
Response 1: We have accepted your suggestion to unify it into the past perfect tense to meet the standard requirements of scientific papers. We comprehensively revised this section to ensure the consistency of tenses, which enhanced the professionalism and rigor of the article.
|
||
5. Additional clarifications |
||
Dear reviewer 1, during the revision process, we added the funding information "the National Natural Science Foundation of China Regional Science Foundation Project (32460817)". This project provided important financial support for sample collection, experimental equipment, and data analysis in our research. Please let us know if you need further information about this project.
|